# Establishment of a Disaster Management-like System for COVID-19 Patients Requiring Veno-Venous Extracorporeal Membrane Oxygenation in Japan

**DOI:** 10.3390/membranes11080625

**Published:** 2021-08-14

**Authors:** Takayuki Ogura, Shinichiro Ohshimo, Keibun Liu, Yoshiaki Iwashita, Satoru Hashimoto, Shinhiro Takeda

**Affiliations:** 1Department of Emergency Medicine and Critical Care Medicine, Tochigi Prefectural Emergency and Critical Care Center, Imperial Gift Foundation SAISEIKAI, Utsunomiya Hospital, Tochigi 321-0974, Japan; 2Department of Emergency and Critical Care Medicine, Graduate School of Biomedical and Health Sciences, Hiroshima University, Hiroshima 734-8551, Japan; ohshimos@hiroshima-u.ac.jp; 3Critical Care Research Group, The Prince Charles Hospital, Brisbane 4032, Australia; keiliu0406@gmail.com; 4Department of Emergency and Critical Care Medicine, Shimane University, Shimane 693-8501, Japan; iwaci1ta@med.shimane-u.ac.jp; 5Department of Anesthesiology and Intensive Care, Kyoto Prefectural University of Medicine, Kyoto 602-8566, Japan; satoru@koto.kpu-m.ac.jp; 6Kawaguchi Cardiovascular and Respiratory Hospital, Saitama 333-0842, Japan; shinhiro@nms.ac.jp

**Keywords:** acute respiratory distress syndrome, coronavirus disease 2019, extracorporeal membrane oxygenation, disaster management, influenza, intensive care unit

## Abstract

The coronavirus disease 2019 (COVID-19) pandemic has increased the number of patients who require extracorporeal membrane oxygenation (ECMO). To manage the demand for ECMO, Japan ECMOnet for COVID-19 was developed as a “disaster management-like system”, utilizing the Cross ICU Searchable Information System (CRISIS) database. This study investigated the effect of the establishment of this disaster management-like system in Japan. This was a nationwide retrospective observational study conducted from 1 February to 31 July in 2020. A total of 187 patients with COVID-19 who received ECMO were included. The median age was 60 years (interquartile range, 53–68), the median length of ventilatory support before ECMO was 3 days (1–5), and the median PaO_2_ to FiO_2_ ratio at ECMO initiation was 86 (71.3–101.5). During the study period, 165 telephone consultations were conducted, including general questions about ECMO. Among them, 44 concerned patients who were already on ECMO or who ultimately received ECMO. Further coordination, including transport and ECMO physician dispatch, was provided for 23 cases. Overall, 125/187 (66.8%) patients were successfully weaned from ECMO. This study demonstrated that Japan has achieved favorable survival outcomes for patients with COVID-19 who received ECMO with a disaster management-like system. Further research on the causes of these outcomes is needed.

## 1. Introduction

In 2020, the coronavirus disease 2019 (COVID-19) pandemic spread from Wuhan, China, throughout the world [1]. In Japan, more than 30,000 patients were diagnosed with COVID-19 by the end of July 2020 [2]; extracorporeal membrane oxygenation (ECMO) was needed for many of these patients who developed severe acute respiratory failure. During the 2009 H1N1 influenza pandemic in Japan, there was a low survival rate for patients with influenza who required ECMO. Takeda et al. stated that the reasons for this low survival rate were the lack of a centralized system and the paucity of respiratory ECMO devices [3]. After the 2009 H1N1 pandemic, academic societies began recommending the use of devices capable of performing respiratory ECMO, as well as dedicated clinician training in the use of such devices. As a result, Japan has achieved improved outcomes for influenza patients treated with ECMO in the non-pandemic era [4]. However, by 2019, a nationwide centralized system for managing the demand for ECMO had not yet been established.

Japan has experienced many natural disasters in recent years and has developed a nationwide disaster management system, including the establishment of Disaster Medical Assistance Teams (DMATs) [5]. As COVID-19 has spread, the need for ECMO has exceeded the available ECMO capacity; therefore, a nationwide “disaster management-like system” for patients with COVID-19 who are possible candidates for ECMO was developed. This system was created by a voluntary group of ECMO experts and was named Japan ECMOnet for COVID-19 (J-ECMOnet). Subsequently, academic societies and the Japanese government have led the group [6]. This investigation aimed to describe the preliminary outcomes of this disaster management-like system for patients with COVID-19 who require ECMO in Japan.

## 2. Materials and Methods

### 2.1. Design and Patients

This was a nationwide retrospective observational study. Consecutive patients with COVID-19 of all ages who were treated with ECMO in Japan and who were registered in the Cross ICU Searchable Information System (CRISIS) database were included. The study period included patients who received mechanical ventilation between 1 February and 31 July 2020. The diagnosis of COVID-19 was confirmed by SARS-CoV-2 real-time reverse transcriptase–polymerase chain reaction from sputum or nasopharyngeal swab samples. Patients whose initial ECMO configuration was venoarterial or veno-arterio-venous were excluded. Publicly available information was obtained from the CRISIS database and included patient age, sex, body mass index, pre-ECMO PaO_2_/FiO_2_ (P/F) ratio, pre-ECMO positive end-expiratory pressure, the number of ventilatory days before ECMO, and the outcome of ECMO treatment (weaned, deceased while on ECMO, still on ECMO). This study used only publicly released data; therefore, institutional review board approval and the need for patient consent were waived. The ethics committees of the Japanese Society of Intensive Care Medicine, the Japanese Association for Acute Medicine, and the Japanese Society of Respiratory Care Medicine agreed to this waiver.

### 2.2. CRISIS Database

The details of the CRISIS database have been described elsewhere [7]. CRISIS is led by the group Japan ECMOnet for COVID-19. More than 600 leading hospitals participate in this system, covering approximately 80% of the intensive care unit (ICU) beds in Japan. The data are manually entered into the CRISIS database by designated staff members from each hospital.

### 2.3. Patient Management

Patients with COVID-19 were treated at the respective hospitals; patients were registered in the CRISIS database if they were transferred to the ICU. The treatment options for each patient varied slightly among the hospitals. Japan has a compulsory insurance system, according to which all those living in Japan must be covered by some form of public insurance. The insurance covers all treatment costs, including ECMO. 

### 2.4. Clinical ECMO Management

J-ECMOnet developed general recommendations regarding the use of ECMO for patients with COVID-19 based on expert opinion. These recommendations were provided by email or website postings to physicians belonging to academic societies for respiratory diseases. Transport of the ECMO patients is divided into two categories: primary transport and secondary transport. Primary transport means that ECMO specialists visit the referring hospital and introduce the ECMO. After receiving ECMO, the patient is transferred to the ECMO center. Secondary transport means patients who have already received ECMO at the referring hospital are transported to the ECMO center.

### 2.5. Japan ECMOnet for COVID-19

The details of J-ECMOnet are described elsewhere [6]. Briefly, J-ECMOnet for COVID-19 is a nonprofit organization and is funded by the government. J-ECMOnet provides the following services:

1. Telephone/e-mail consultation for physicians.

General educational information about respiratory ECMO is provided every day, 24 h per day, 365 days per year, by telephone and e-mail. Experienced members of J-ECMOnet provide information to physicians.

2. Database development.

We developed the CRISIS system, which is a real-time, web-based, information-sharing system for ICU beds available in Japan. The database includes ICU beds in 620 hospitals (87.9% coverage) in Japan. Registration in the CRISIS database is mainly completed by local hospitals. This database is regularly summarized, and the information is made publicly available on the internet. Additionally, a limited version of the database is publicly available and provides real-time information on which hospitals have beds to accommodate COVID-19 patients with severe respiratory failure. This simplified dataset can be accessed from the CRISIS website.

3. ECMO expert dispatch.

ECMO expert dispatch is occasionally activated when a regional coordinator determines that it is too difficult to establish an appropriate treatment plan solely by telephone and/or e-mail or if a more detailed discussion is necessary to reach an appropriate treatment decision. Experts provide on-site ECMO cannulation and coaching services at local hospitals to maintain qualified ECMO care.

4. Organize primary or secondary ECMO transport services.

If a regional coordinator determines that a patient requires transportation, then they will coordinate transportation with the referring hospital. If a suitable hospital cannot be found nearby, then the regional coordinator consults with the general coordinator to find a suitable hospital in the wider region.

### 2.6. Disaster Management-Like System for ECMO

A flowchart describing the J-ECMOnet system is shown in Figure 1.

Japan ECMOnet for COVD-19 provided ECMO coordination and support. Regional ECMO coordinators were widely positioned throughout the country—71.3% of Japan’s population was covered by these coordinators. Coverage was defined as the ability of responders to arrive at a referring hospital within 2–3 h of calling. When requests for ECMO coordination were received by telephone [A], regional ECMO coordinators dispatched a Rapid Response ECMO Team to the hospital to initiate ECMO [B-1]. Coordinators also identified available ECMO beds and arranged transportation for patients while on ECMO [B-2]. If the regional coordinator found no available ECMO beds near the referring hospitals, the general coordinator searched for such beds throughout Japan [C]. If no available ECMO bed was available in Japan at that time, the team stayed at the local hospital and provided on-site ECMO coaching for several days [B-3] and continued remote ECMO management after leaving [B-4].

Recommendations are provided upon receipt of telephone or e-mail consultation requests. ECMO consultants are available 24 h per day for telephone consultation [6]. The recommendations emphasize the following six key points for providing ECMO care to patients with COVID-19:Before initiating ECMO, ventilator settings should be consistent with a lung-protective strategy.Physicians should be aware that the use of mechanical ventilation with high levels of support for 7 days or more before initiating ECMO increases the risk of mortality [8].Consider initiating ECMO for patients with COVID-19 receiving positive end-expiratory pressure >10 cmH_2_O and who have PaO_2_/FiO_2_ < 100 and a deteriorating respiratory condition.ECMO should only be provided for patients with reversible lung injury [8].The use of ECMO for older patients (e.g., 75 years or older) is generally associated with poor outcomes (however, there is no specific age contraindication) [9].Physicians should consult J-ECMOnet at any time if they are concerned about providing ECMO support for patients with COVID-19.

Notably, no specific drugs or therapies for patients with COVID-19 are mentioned in these recommendations.

The ECMO coordination and support protocols carried out by J-ECMOnet are similar to the disaster management protocols carried out by DMATs [5]. Regional and general ECMO coordinators are located in strategic regions throughout the country. Overall, 71.3% of Japan’s population is covered by these coordinators (Figure 2), with coverage being defined as the ability of responders to arrive at a referring hospital within 2–3 h of receiving a consultation request.

The stars indicate the locations of the ECMO coordinators and ECMO teams. The circles indicate the locations of ECMO consultants (i.e., those who are not part of a larger ECMO team). The number in each area indicates how many patients underwent ECMO during the study period.

When requests for ECMO coordination are received from ECMO consultants, the regional ECMO coordinator in charge of the area decides how to best manage the patient. The tasks of ECMO coordinators include identifying available ECMO beds near the referring hospital, ensuring that vehicles are prepared to transport the patient while undergoing ECMO, and deciding whether to dispatch an ECMO expert to the referring hospital. If the regional coordinator cannot find an available ECMO bed near the referring hospital, then the general coordinator searches throughout Japan for an available ECMO bed. Every search for an ECMO bed is completed using CRISIS.

### 2.7. Statistical Analysis

Continuous variables are presented as the median and interquartile range (IQR), and categorical variables are expressed as values with percentages. Survival analysis was carried out using Kaplan–Meier survival estimates, and differences between the survival curves were compared by log–rank analysis. For data availability reasons, we used the date of tracheal intubation through the date of tracheal extubation to generate the survival curve of COVID-19 patients. Similarly, the date of hospitalization through the last survival confirmation date was used for influenza patients. Detailed data on influenza patients are available in another article [4]. All statistical analyses were conducted using SPSS software version 22.0 for Windows (IBM Corp, Armonk, NY, USA).

## 3. Results

A total of 187 patients with COVID-19 who received ECMO support were included in this study. The median age was 60 years (IQR, 53–68), 156 (83.4%) were men, the median length of time undergoing ventilatory support before ECMO was 3 days (1–5), and the median P/F ratio upon ECMO initiation was 86 (71.3–101.5) (Table 1). In comparison with the flu of 2009 and the flu of 2016, the age of the patients was higher, PEEP at the initiation of ECMO was lower, and PaO_2_/FiO_2_ at the initiation of ECMO was higher.

ECMO: extracorporeal membrane oxygenation; PEEP: positive end-expiratory pressure; PaO_2_: partial pressure of oxygen; F_i_O_2_: fraction of inspiratory oxygen; IQR: interquartile range

During the study period, a total of 165 telephone consultations were conducted by J-ECMOnet. These included actual case consultations as well as consultations about the general preparation of ECMO settings. Overall, 44 (26.7%) of the telephone consultations were about patients who were already on ECMO or who ultimately received ECMO. Primary ECMO transport was conducted for five patients, secondary ECMO transport was conducted for four patients, conventional transport was conducted for ten patients, and ECMO physician dispatch was conducted for four patients (Table 2). Once transportation was planned, J-ECMOnet coordinated the operation. In every operation, well-trained experts were deployed from J-ECMOnet to ensure safe transportation. There were no significant adverse events associated with the transportations.

In total, 125/187 (66.8%) of the patients were successfully weaned from ECMO, while there were 62/187 (33.1%) deaths. The survival curve of the included patients is shown in Figure 3 in comparison with the survival curves of patients with H1N1 influenza in 2009 and 2016 [3,4]. The median follow-up period was 24 days (15–39). The survival rate of patients with COVID-19 was significantly higher than that of patients with H1N1 influenza in 2009 (*p* < 0.010), but was not significantly different from that of patients with H1N1 influenza in 2016 (*p* = 0.11).

A total of 187 patients with COVID-19 who underwent ECMO were included in this study. The COVID-19 survival curve was significantly higher than the 2009 H1N1 influenza curve (*p* = 0.010), but not significantly different from the 2016 H1N1 influenza curve (*p* = 0.11).

## 4. Discussion

In this descriptive study, we report the establishment of a treatment network for patients with severe COVID-19 in Japan and its outcomes in the first half of 2020. A total of 187 patients were registered in the CRISIS database. Overall, 66.8% of these patients were successfully weaned from ECMO, suggesting that the Japanese treatment network for COVID-19 patients and the cooperative system of telephone, e-mail, and on-site visits may have contributed to the favorable outcomes of these severe COVID-19 patients.

The establishment of a nationwide disaster management-like system that utilizes the CRISIS database, a telephone consultation system, and ECMO coordination has enabled Japan to achieve better ECMO outcomes during the COVID-19 pandemic than during the H1N1 pandemic. In the H1N1 influenza pandemic, Japan had a low ECMO patient survival rate (35%) [3]. This poor outcome has been attributed to insufficient technical knowledge of ECMO, the lack of available respiratory ECMO devices, and the absence of a national centralized system for managing these special cases. A study from 2016 (a non-pandemic year) showed the improved survival outcomes of influenza patients who required ECMO, which was attributed to the increasing availability of appropriate devices [4]. However, a nationwide centralized system in Japan had still not been constructed as of 2019. Therefore, the survival rates of ECMO patients in a pandemic year were unclear.

The survival outcomes found in our study are similar to those of international studies. Initial reports from China showed a 94.1% mortality for COVID-19 patients treated with ECMO [10]. This high mortality rate was probably because COVID-19 was a newly emerging disease and its natural course was unknown. A study from Italy showed that 39/71 (54.9%) patients with COVID-19 died while on ECMO, while 6/71 (8.5%) died after ECMO removal. The mean P/F ratio before ECMO was 78.7 ± 39.3, the time between ICU admission and ECMO insertion averaged 11.6 ± 8.9 days, and the mean pre-ECMO intubation time was 6.5 ± 5.3 days [11]. Our results show an average of 3 days before the initiation of ECMO, which differs from that in the Italian study. A contributing factor might be that although Italy experienced a large wave of patients with COVID-19, this was not the case in Japan. A recent study in France showed a 60-day mortality rate of 31% and a median pre-ECMO P/F ratio of 60 [12]. A US study showed a median patient age of 49 years, a pre-ECMO P/F ratio of 72, and a 60-day mortality of 33.2% (63/190) [13]. Our previous report showed that 16/26 (62%) patients were successfully weaned from ECMO [14] after the establishment of a disaster management-like system for ECMO usage; in the current study, 125/187 (66.8%) patients were successfully weaned. Although the included data did not contain the details of each patient, the ranges of patient age, sex, and pre-ECMO P/F ratio were similar to those in the France and US studies. Therefore, the patients’ background characteristics can be considered as similar to those found in the France and US studies. However, additional analysis that includes the patients’ disease severity, ventilatory status, and ECMO management details are needed to provide a more thorough comparison.

We considered this pandemic to be a disaster. In this context, a disaster was defined as “medical needs that exceed the medical supply”. Therefore, we managed the provision of ECMO treatment similarly to a disaster management response. In Japan, despite the increasing number of available ECMO machines, there are few experienced respiratory ECMO centers. There is also no government-led centralized system for managing the provision of this specialized care. The increasing need for ECMO care might lead to less experienced centers attempting to treat these patients on their own, which may result in suboptimal outcomes. Japan is a unique country that experiences many natural disasters; therefore, a national disaster management system has been constructed. We introduced a nationwide ECMO management system that is analogous to the disaster management system. In a disaster, medical staff must work systematically to achieve the greatest benefit for patients. In disaster situations in Japan, the local DMATs (first responders) are dispatched to the “hot zone”, while the DMAT headquarter office commands the local team [5]. During the COVID-19 pandemic, J-ECMOnet played an important role in coordinating with the government. Analogous to the DMAT command structure, J-ECMOnet utilized a similar command system to manage the provision of ECMO by ensuring that policies for the use of ECMO are adhered to and by directing the dispatch of ECMO experts to local hospitals. The development of this command system, with limited ECMO experts, enabled the successful coordination of ECMO treatment, patient transport, on-site coaching, and remote care by staff from the referring hospital. The response of J-ECMOnet to the substantial need for ECMO may have contributed to the positive patient outcomes. This disaster-like centralized regional ECMO coordination system enabled the appropriate distribution of ECMO resources. Coordinators determined whether patients could be treated at local hospitals or whether they required transport to larger facilities. This approach prevented the overload of hospitals with specialty ECMO services.

There were three major limitations to this study. First, detailed clinical information was lacking (e.g., the severity of respiratory impairment, comorbidities, the treatment provided, and ICU staffing levels). This is because CRISIS was originally developed as a public information system that collects clinical summaries of patients treated in ICUs and the hospitals’ ICU bed capacity. Therefore, it is not clear whether the positive outcomes are due to the J-ECMOnet system or because the overall disease severity was low. However, the patient age, sex, and pre-ECMO P/F ratio distributions were similar to those found in previous studies, indicating a similar disease severity. Second, this was a retrospective observational study; therefore, it is not clear whether the outcomes were owing to the centralized system or to other factors. Third, the follow-up was relatively short, and the mortality reported in this study could change. Continued data collection is important to obtain a more accurate evaluation of ECMO use for patients with COVID-19 in Japan.

## 5. Conclusions

In conclusion, this nationwide retrospective observational study found a relatively favorable survival rate for patients with COVID-19 who underwent ECMO in Japan. We believe that the development of the nationwide disaster management-like system contributed to this positive outcome. Further research on the causes of these outcomes is needed.

## Figures and Tables

**Figure 1 membranes-11-00625-f001:**
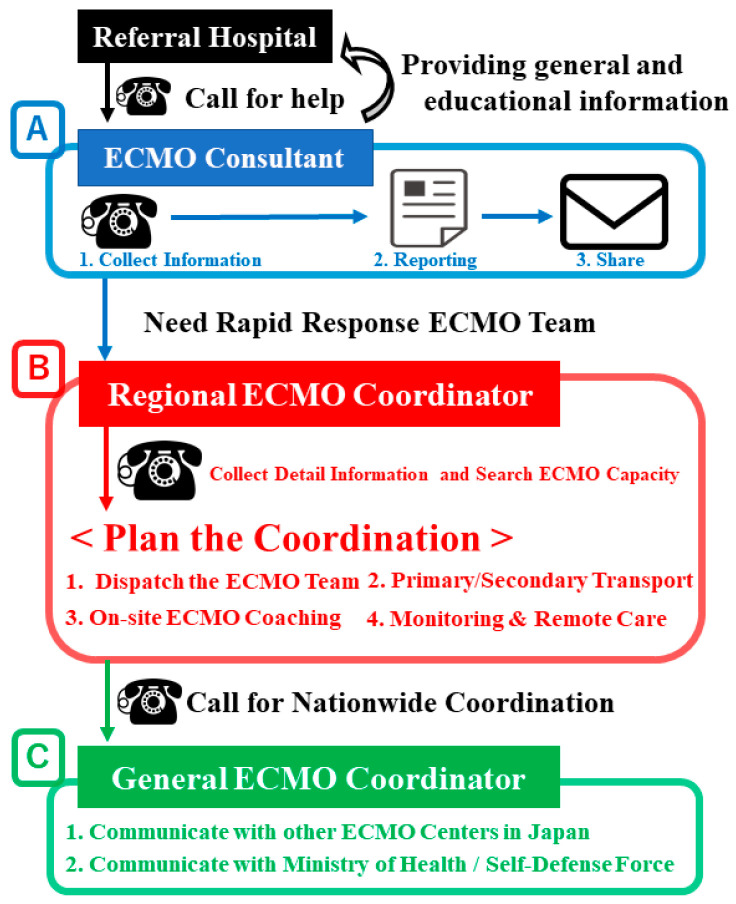
Flowchart of the Japan ECMOnet operation and ECMO coordinator deployment.

**Figure 2 membranes-11-00625-f002:**
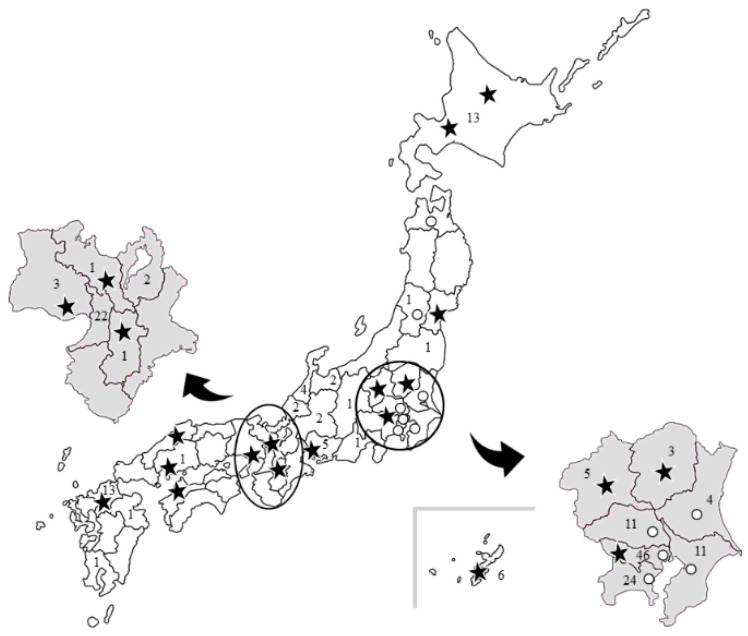
Distribution of ECMO coordinators and the number of patients with COVID-19 undergoing ECMO in Japan.

**Figure 3 membranes-11-00625-f003:**
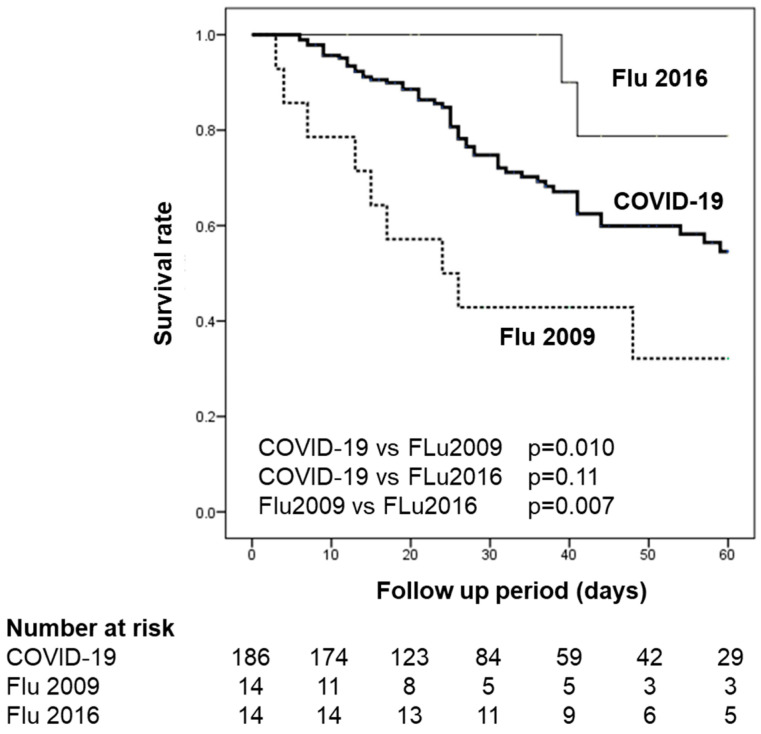
Kaplan–Meier survival estimates during the first 60 days of the observational study.

**Table 1 membranes-11-00625-t001:** Patient characteristics.

	Flu 2009	Flu 2016	COVID-19	*p*-Value
Number of patients	14	14	187	
Age (years, IQR)	54 (43–60)	52 (43–63)	60 (53–68)	0.0040
Gender (male, %)	12 (85.7%)	12 (85.7%)	156 (83.4%)	0.97
Length of time of ventilatory support before ECMO (days, IQR)	5.0 (1.0–7.0)	1.0 (1.0–2.8)	3.0 (1.0–5.0)	0.12
PEEP at the initiation of ECMO (cmH_2_O, IQR)	24 (19–30)	15 (14–19) *	12 (10–15) *	<0.001
PaO_2_/F_I_O_2_ at the initiation of ECMO (number, IQR)	50 (41–53)	70 (58–75) *	86 (71–102) *	<0.001

* Lowest PEEP and PaO_2_/F_I_O_2_ before initiating ECMO.

**Table 2 membranes-11-00625-t002:** Services provided (cumulative number).

Telephone-based consultation	165
ECMO coordination	54
Activation of the Rapid Response ECMO Team	11
On-site coaching or remote care	24
ECMO transport	6

ECMO: extracorporeal membrane oxygenation.

## Data Availability

Not applicable.

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
