# Peer review of "Establishment of a Disaster Management-like System for COVID-19 Patients Requiring Veno-Venous Extracorporeal Membrane Oxygenation in Japan"

_membranes, 2021, doi:10.3390/membranes11080625_

Round 1

Reviewer 1 Report

  1. If possible, authors would be better to describe data of 2016 Flu and 2009 Flu seasons at least compatible to COVID-19.
  2. Describe the process of transportation COVID-19 critically ill patients under ECMO. Transportation ECMO patients have been already known to be very difficult and risky, so I wonder if there is something bad or wrong situation in the real world.
  3. Please edit the tables.

Author Response

Reviewer 1

  1. If possible, authors would be better to describe data of 2016 Flu and 2009 Flu seasons at least compatible to COVID-19.

We have added the data of 2016 Flu and 2009 in table 1.

  1. Describe the process of transportation COVID-19 critically ill patients under ECMO. Transportation ECMO patients have been already known to be very difficult and risky, so I wonder if there is something bad or wrong situation in the real world.

Thank you so much for pointing out this. It is true that transportations of patient on ECMO are at high risk, especially for inexperienced facilities. Thus, once transportation was planned, J-ECMOnet coordinated the operation. In every operations, well trained experts were deployed from J-ECMOnet to support for safe transportation. There were no significant adverse events associated with transportations. We have added this information inLine 220.

  1. Please edit the tables.

We have edited the tables.

Reviewer 2 Report

The authors present a very interesting and unique perspective on the rapid nationwide deployment of a crisis management system for ECMO utilization in the face of a pandemic. The study is well designed, easy to follow and adds significantly to the literature. It provides a model that could be adapted for similar situations and in other countries.

There are several areas that could strengthen the current manuscript. The CRISIS management system and J-ECMOnet are unique enough that they should be the focus of the manuscript with more detailed description of the different aspects of that experience as in time to deployment of teams to aid local hospitals, time spent by local teams at the deployed hospitals to aid in ECMO management when transport was not possible, number of times transport was recommended but not possible and such.

The comparison to the 2009 and 2016 data is unclear. There is no clear description for the source of the 2009 and 2016 data, especially as the 2020 data was developed for the current pandemic specifically so the system was not there to collect data in 2009 or 2016. Additionally, comparing outcomes to either of those eras, especially in the face of improved technology and gained experience with ECMO use is distracting and difficult to interpret.

The authors attribute the improved outcomes to the current crisis management system, a conclusion that cannot be made. While the disaster management system is unique and clearly influenced patient management, there is no evidence for causality. I would greatly caution the authors from making such conclusions.

Author Response

Thank you so much for your favorable comments.

The comparison to the 2009 and 2016 data is unclear. There is no clear description for the source of the 2009 and 2016 data, especially as the 2020 data was developed for the current pandemic specifically so the system was not there to collect data in 2009 or 2016. Additionally, comparing outcomes to either of those eras, especially in the face of improved technology and gained experience with ECMO use is distracting and difficult to interpret.

-->The data of 2009 and 2016 are referred to reference No 3 and 4 those are focused on H1N1 influenza. The detailed basic characteristics are added to Table 1. 

The authors attribute the improved outcomes to the current crisis management system, a conclusion that cannot be made. While the disaster management system is unique and clearly influenced patient management, there is no evidence for causality. I would greatly caution the authors from making such conclusions.

-->We have changed the description. This study demonstrated that Japan has achieved favorable survival outcomes for patients with COVID-19 who received ECMO with disaster management-like system. Further research on causality of this outcomes are needed.

Reviewer 3 Report

Ogura et al. present a very interesting approach to how a pandemic should be addressed. By centralizing information and concentrating experts on a field with widespread accessibility but not - so widespread know-how of the users, it is paramount to be able to support physicians in need of advice. This can lead to a better treatment for our patients throughout entire countries, without the need of a special expert in every center. I found the concept and the presentation very interesting and of clinical use, this should be done also in other countries.
I have only minor comments that need to be addressed:

  • line 28: "This study demonstrated that the 
    29 Japan ECMOnet disaster management-like system achieved favorable survival outcomes..." is a little overstated. Unfortunately, this can be only stated if there is a randomized control group. The study demonstrated that it is feasible, that is reasonable, that it can be useful. Please edit accordingly in abstract and text.
  • line 198 : Please cite SPSS the right way, it's not Chicago (https://www.ibm.com/support/pages/how-cite-ibm-spss-statistics-or-earlier-versions-spss)
  • Please add the Numbers at risk table to the Kaplan meier graph
  • line 264: i'm not sure the cited italian study has the right citation 11. please double - check.
  • I may have missed something. But how is ECMOnet financed? Who is running it and the infrastructure?

Author Response

Reviewer 3

Ogura et al. present a very interesting approach to how a pandemic should be addressed. By centralizing information and concentrating experts on a field with widespread accessibility but not - so widespread know-how of the users, it is paramount to be able to support physicians in need of advice. This can lead to a better treatment for our patients throughout entire countries, without the need of a special expert in every center. I found the concept and the presentation very interesting and of clinical use, this should be done also in other countries. I have only minor comments that need to be addressed:

  • line 28: "This study demonstrated that the 29 Japan ECMOnet disaster management-like system achieved favorable survival outcomes..." is a little overstated. Unfortunately, this can be only stated if there is a randomized control group. The study demonstrated that it is feasible, that is reasonable, that it can be useful. Please edit accordingly in abstract and text.

We have changed the description. This study demonstrated that Japan has achieved favorable survival outcomes for patients with COVID-19 who received ECMO with disaster management-like system. Further research on causality of this outcomes are needed.

  • line 198 : Please cite SPSS the right way, it's not Chicago (https://www.ibm.com/support/pages/how-cite-ibm-spss-statistics-or-earlier-versions-spss)

We have changed the description as follow; IBM Corp, Armonk, NY

  • Please add the Numbers at risk table to the Kaplan meier graph

We have added the Numbers at risk table in Figure 3.

  • line 264: i'm not sure the cited italian study has the right citation 11. please double - check.

We have edited the reference list.

  • I may have missed something. But how is ECMOnet financed? Who is running it and the infrastructure?

⇨Japan ECMOnet for COVID-19 is a nonprofit organization funded for COVID-19 pandemic. It is funded by Government. We have added this information in Line 102